# Cross-Cultural Adaption and Validation of the Dutch Version of the Kerlan-Jobe Orthopaedic Clinic Questionnaire in Juvenile Baseball Pitchers

**DOI:** 10.3390/sports10110163

**Published:** 2022-10-24

**Authors:** A. J. R. Leenen, Amber Hurry, Femke van Dis, Erik van der Graaff, H. E. J. Veeger, M. J. M. Hoozemans

**Affiliations:** 1Department of Human Movement Sciences, Faculty of Behavioural and Movement Sciences, Amsterdam Movement Sciences, Vrije Universiteit Amsterdam, 1081 BT Amsterdam, The Netherlands; 2Fysiokliniek Amsterdam, 1067 SM Amsterdam, The Netherlands; 3PitchPerfect, 4814 GA Breda, The Netherlands; 4Department of Biomechanical Engineering, Faculty of Mechanical, Maritime and Material Engineering, Delft University of Technology, 2628 CD Delft, The Netherlands

**Keywords:** overhead athlete, patient-reported outcome, questionnaire, cross-cultural validation study, upper-extremity, physical function

## Abstract

Monitoring the performance and functional status of baseball pitchers’ upper extremity is important in maintaining the athlete’s health and performance. This study validated a Dutch translation of the original English Kerlan-Jobe Orthopaedic Clinic (KJOC) against the previously validated Disabilities of the Arm, Shoulder and Hand (DASH) and Western Ontario Shoulder Instability Index (WOSI) questionnaires in a group of talented juvenile Dutch baseball pitchers. Three times, from 2014–2016, 107 pitchers completed the Dutch KJOC, DASH and WOSI questionnaires. Participants’ questionnaire scores were analysed for the whole group and the symptomatic player subgroup separately. Internal consistency, construct validity and ceiling and floor effects were examined. Cronbach’s alpha was consistently above 0.8 for the three time periods for the whole group, and ranged between 0.62 and 0.86 for the symptomatic subgroup. Spearman’s rank correlation coefficients ranged from 0.47 to 0.67 for the whole group and 0.32 to 0.99 for the symptomatic subgroup. No floor effects were observed in the scores of the KJOC and only a ceiling effect for the whole group (15.2%) at one time period. The Dutch version of the KJOC has shown acceptable internal consistency and construct validity and can be used to assess overhead athletes’ shoulder and elbow functionality.

## 1. Introduction

Overhead sports athletes are put at increased risk for the development of upper extremity overuse injuries due to the repetitive and explosive nature of the practiced motions, such as when throwing a baseball [1,2]. Self-report questionnaires are a useful tool to evaluate the functional status and performance of these overhead athletes. In addition, self-report questionnaires allow for subjective thoughts and beliefs to be quantified and evaluated using a standardized procedure. Previously developed questionnaires are designed to evaluate self-reported upper extremity function or performance for non-athletes or athletes who do not specifically participate in overhead sports [3,4]. However, most of these frequently used questionnaires, such as the Disabilities of the Arm, Shoulder and Hand Questionnaire (DASH), Western Ontario Shoulder Instability Index (WOSI) and American Shoulder and Elbow Surgeons Standardized Shoulder Assessment Form (ASES), are known for their ceiling effect in high-performance overhead sports, limiting the ability to detect subtle sports-related changes in upper extremity function or performance [5,6,7].

One questionnaire that is suitable to evaluate the self-reported functional status of the upper extremity of baseball pitchers is the Kerlan-Jobe Orthopaedic Clinic shoulder and elbow questionnaire (KJOC) [8,9]. The KJOC questionnaire, originally developed in English by the Kerlan-Jobe Orthopaedic Clinic, seeks to measure the functional status of the upper extremity in the overhead athlete. The questionnaire features questions regarding upper extremity function and throwing performance [8]. The KJOC is able to discriminate between injured and non-injured overhead athletes, and those competing with arm trouble and not competing due to arm trouble [8,10,11]. The questionnaire has been proven to be a valid, reliable and responsive instrument to evaluate shoulder and elbow injuries in various overhead athletes [8,9,12,13,14].

In the original study by Alberta et al. [8], and all previous cross-cultural adaptation studies, the KJOC questionnaire is validated against the DASH questionnaire, as both questionnaires attempt to map the overall functional status of the upper extremity [8,11,15,16,17,18]. However, baseball pitchers usually experience symptoms in a specific part of the upper extremity rather than the entire upper extremity (Leenen et al. almost submitted). Shoulder instability, for instance, is one of those symptoms that pitchers experience in the shoulder region that regularly negatively affects shoulder function and performance. However, the KJOC questionnaire has never been validated against a disease- and region-specific questionnaire, such as the WOSI, which aims to evaluate quality of life associated with shoulder instability [8,19]. In addition, this WOSI questionnaire is one of the few questionnaires that does not primarily focus on activities that occur in daily life [7].

The KJOC questionnaire was originally written in English and has been translated into many different languages [11,15,16,17,18] (Appendix A). As the exchange of resources across international borders becomes an increasingly common occurrence, it is important that material can be successfully translated and applied within multicultural settings. Cross-cultural evaluations of questionnaires are important to assess the validity and reliability of the content to ensure that responses can be correctly acquired. The present cross-cultural validation study will focus on the translation of the original English version of the KJOC questionnaire into Dutch and its validity compared to the validated Dutch versions of the DASH and WOSI questionnaires [7,20]. Therefore, the aim of the present study is to develop the Dutch version of the KJOC questionnaire through cross-cultural adaptation and to verify reliability, validity and interpretability in talented juvenile Dutch baseball pitchers.

## 2. Materials and Methods

### 2.1. Translation Procedures

The KJOC questionnaire was translated according to the guidelines outlined by Beaton et al. [21]. Following these guidelines ensured that the questionnaire was effectively translated, making the questionnaires linguistically correct and conceptually accurate. For this research, the original English version of the questionnaire was translated into Dutch with the help of two native Dutch speakers. One of these translators was aware of the study background, while the other was not briefed on the research and had no medical background. This version was then translated back from Dutch into English by two native Dutch individuals with sufficient command of the English language. Both individuals came from a non-medical background and were not briefed before the translation. The final translation that produced the Dutch version of the KJOC was conducted by an expert panel comprising translators, researchers and healthcare professionals (Appendix B).

### 2.2. Study Design and Study Population

In a two-and-a-half year prospective, dynamic cohort study, participants comprised talented juvenile Dutch baseball pitchers aged 12–18 years, who participated in one of the six Dutch regional baseball talent academies and the Dutch National U-18 team in the seasons 2014–2016. In total, 107 talented juvenile Dutch baseball pitchers participated in this study. Demographics of the participants are listed in Table 1. Between the six test moments, players could leave and enter the study due to, for instance, age, (de)selection, injury or recovery, and giving or withdrawing consent to participate in the study. The participants filled out questionnaires at the beginning and end of the baseball season for three consecutive seasons (i.e., March 2014, October 2014, March 2015, October 2015, March 2016, October 2016). In the present validation study, data from the first three consecutive measurements were used for analysis. At the first measurement (T1), 87 participants completed the questionnaires, followed by 79 participants at the second and third measurements (T2 and T3), which showed a 9.2% drop-out rate. The validation study surpassed the 50-participant minimum required to meet the guidelines set by Terwee et al. [22] for the appropriate analysis of questionnaires measuring an individual’s health status. The study was approved by the Faculty of Behavioural and Movement Sciences’ local ethics committee of the Vrije Universiteit Amsterdam (protocol number ECB-2013-53), and all participants or their legal representatives gave their written consent according to the university policy after being fully informed about the content and purpose of the study. 

### 2.3. Procedure and Data Collection

The Dutch versions of the KJOC and DASH questionnaires were completed by all participants at their local training facility at each of the three measurements (in March 2014 (T1), October 2014 (T2) and March 2015 (T3)). Participants who required ‘medical attention’ or missed game or practice time in the last six months due to upper extremity symptoms were classified as symptomatic players (subgroup part of the whole group). The participants that were classified under the heading of ‘medical attention’ were those who had consulted a (para)medic care provider in the last six months due to experienced upper extremity symptoms. The participants who in any case experienced shoulder symptoms in the last six months were asked to also fill out the Dutch version of the WOSI questionnaire [23]. Finally, they filled out an accompanying general questionnaire concerning their age, body height and body mass (and body mass index [BMI] was calculated).

### 2.4. Questionnaires

The KJOC questionnaire consists of ten items scored with visual analogue scales (VAS) ranging from 0 to 100 millimetres. It includes the two subscales function (5 items) and performance (5 items) to evaluate the shoulder and elbow function, performance and pain in overhead athletes [8]. Participants answered the items according the score being produced from the mark placed on the VAS. The score was expressed to one decimal point (e.g., a score of 70 mm on one question was expressed as 7.0 of 10). The maximum score a participant could achieve on each item of the questionnaire was 10. The unweighted summed score of all items corresponded to 100 points, representing the best possible shoulder and elbow function and performance. Up to two missing items were tolerated within the KJOC questionnaire responses, and then the remaining scores were averaged to produce the final average score for each period (T1, T2 or T3).

The 30-item DASH questionnaire was designed with subjective 5-point Likert scales, ranging from no difficulty to unable, from none to extreme, or from no impact to high impact. Three subscales were included: physical function (21 items), symptoms (6 items), and the subscale social or role function (3 items) [6]. The lowest sum score of 0 points corresponds with a minimal disability and the highest possible sum score of 100 indicates maximal disability [6]. DASH questionnaire responses were excluded if more than 3 items were missing.

The WOSI questionnaire consists of a 21-item VAS ranging from 0 to 100 mm, including the domains physical symptoms and pain (10 items), sports, recreation and work (4 items), lifestyle and social functioning (4 items) and emotional well-being (3 items), to evaluate shoulder symptoms experienced by overhead athletes [7]. The maximal unweighted summed score of 2100 signifies the worst shoulder-related quality of life relative to the lowest score of 0. As with the DASH questionnaire, answers to the WOSI questionnaire were excluded if more than 3 items were missing.

### 2.5. Psychometric Properties

#### 2.5.1. Internal Consistency

The internal consistency, as a measure of the homogeneity of the ten items of the KJOC, was evaluated using Cronbach’s alpha [24]. The internal consistency with corresponding 95% confidence intervals (CI) was determined at each time period. A Cronbach’s alpha value above 0.7 is widely considered a measure of acceptable internal consistency, indicating high correlations among the items within the scale, while values below 0.7 indicate poor internal consistency [25].

#### 2.5.2. Construct Validity

The construct validity of the KJOC was evaluated by determining Spearman’s rank correlation of the KJOC scores with both the DASH and WOSI scores. A Spearman’s rho < 0.39 was considered a weak correlation, 0.40–0.69 a moderate correlation, 0.70–0.89 a strong correlation, and >0.90 showed a very strong correlation [26].

#### 2.5.3. Interpretability

Interpretability is considered an important characteristic of a measurement instrument that refers to the degree to which qualitative meaning can be assigned to the quantitative scores of an instrument [27]. One aspect of interpretability is assessing floor and ceiling effects. Floor and ceiling effects were present as instances whereby 15% or more of the participants obtained the highest or lowest score [28]. The highest KJOC score corresponding to a score of 100 and the lowest DASH and WOSI score corresponding to a zero score were considered a floor effect. Thus, the lowest KJOC and highest DASH and WOSI scores were considered a ceiling effect.

### 2.6. Statistical Analysis

Statistical analyses were separately performed for each time period (T1, T2 and T3) and questionnaire score. The symptomatic players were assumed to score lower on the KJOC compared to the whole group and to score lower on average on each time period. Therefore, statistical analyses were performed separately on the whole group and on the group of symptomatic pitchers, as it was postulated that the KJOC within this symptomatic subgroup would exhibit greater consistency and validity than the whole group.

The distributions of the KJOC, DASH and WOSI sum scores underwent separate normality checks for each time period. All distributions, except for the WOSI sum scores, were accompanied with significant Shapiro–Wilk normality tests (*p* < 0.05). Statistical analyses were performed using R (R Core Team, version 4.0.0, 2020, Vienna, Austria) with ggplot2 (version 3.3.2) to design the boxplots [29,30] and an a priori α level of 0.05 was used to determine statistical significance.

## 3. Results

The players that experienced upper extremity symptoms were common during the study, which is consistent with the high injury rates associated with baseball pitching. There were 14 symptomatic players at T1, 6 players at T2 and 7 players at T3 (Table 2). A total of 23 symptomatic players accounted for 21.5% (23 of the 107 participants) of the whole group.

### 3.1. Internal Consistency

The internal consistency of the KJOC questionnaire was consistently high, with a Cronbach’s alpha averaging over 0.80 for all periods apart from the symptomatic players subgroup at measurement period T1, with a value of 0.62 (95% CI [0.20, 0.80]) (Table 3). This value is below the desired value of 0.70. Overall, internal consistency results demonstrate a good to acceptable internal consistency.

### 3.2. Construct Validity

The construct validity scores varied greatly across questionnaires, analysis groups and periods, with Spearman’s rho values ranging from 0.99 for the correlation between KJOC scores and DASH scores (at period T1 in the symptomatic players subgroup), which demonstrates a very strong correlation, to a Spearman’s rho value of 0.32 for the correlation between KJOC scores and WOSI scores (at period T2 in the symptomatic upper extremity players subgroup), which demonstrates a fair correlation (Table 4). Overall, the Spearman’s rho values were higher for the symptomatic players subgroup; therefore, a higher level of construct validity was observed for these KJOC scores in relation to the DASH and WOSI scores.

### 3.3. Interpretability

Overall, no floor effect was observed in the KJOC questionnaire scores, for both the whole group and the symptomatic players subgroup, across T1 and T2, with only T3 showing a 15.2% ceiling effect for the whole group (Table 5).

## 4. Discussion

This study project sought to cross-culturally validate the Dutch translation of the KJOC against the previously validated Dutch versions of the DASH and WOSI questionnaires. Internal consistency, construct validity, and floor and ceiling effects were analysed for a group of juvenile baseball pitchers across three measurement periods, with separate analysis for the pitchers with upper extremity symptoms. The results demonstrated that the Dutch version of the KJOC is a valid tool to assess shoulder and elbow function, performance and pain in talented juvenile baseball pitchers. Floor and ceiling effects for the KJOC questionnaire were minimal.

The internal consistency of the KJOC questionnaire was assessed using Cronbach’s alpha. The results from this study showed a high average Cronbach’s alpha value of 0.84 over all time periods for the whole group, indicating good internal consistency among the 10 items. For the group of pitchers with upper extremity symptoms, similarly high Cronbach’s alpha values were seen, with an average of 0.77 across all time periods, indicating acceptable internal consistency. These results are in accordance with previous studies, who reported good to excellent internal consistency [8,14,15,16,17]. Cronbach’s alpha values were, however, slightly lower in this study, which is likely due to the fact that this study only included baseball pitchers, while the aforementioned studies included players from various overhead sports, such as handball, badminton and basketball. Nevertheless, the internal consistency of the KJOC questionnaire is more than acceptable for application of the questionnaire within the population of talented juvenile baseball pitchers. 

The construct validity of the newly translated Dutch KJOC was assessed by comparing this questionnaire against previously validated Dutch DASH and WOSI questionnaires. While assuming that the DASH and the KJOC questionnaires measure similar constructs, the DASH mainly focuses on activities that occur in daily living, whereas the KJOC questionnaire aims to measure the functional status of the upper extremity in the high functioning population of overhead athletes. The results from this study showed that the averaged Spearman’s rho value was found to be −0.58 across all the periods for the whole group, indicating moderate construct validity. These results are in close agreement with previous studies that examined the construct validity of the KJOC against the DASH in other languages [11,15,16,17]. However, the Spearman’s rho values in this study were slightly lower than those reported in the original study [8]. The study population in the present study consisted of baseball pitchers with and pitchers without upper extremity symptoms, whereas the study of Alberta et al. [8] examined the construct validity of the KJOC questionnaire against the DASH in a study population that consisted of overhead athletes who were free of symptoms. The construct validity was expected to be higher for the more homogeneous symptomatic player subgroup compared to the relatively heterogeneous whole group. Indeed, the Spearman’s rho value averaged over the measurement periods for the symptomatic player group was −0.75, indicating strong construct validity, compared to the value of −0.58 for the whole group, as mentioned above. In contrast, the averaged Spearman’s rho values for the KJOC scores against the WOSI scores over all periods was found to be −0.60, which is slightly lower compared to the KJOC against the DASH. This may be due to the fact that the DASH questionnaire assesses the degree of upper extremity disability in activities of daily living, whereas the disease- and region-specific WOSI questionnaire attempts to assess the quality of life related to shoulder instability. Nevertheless, knowing that these questionnaires do not consider the specific demands of overhead athletes, the KJOC questionnaire is better able to accurately assess upper extremity functional status in this population of juvenile baseball pitchers.

Since the Dutch version of the KJOC questionnaire did not show any floor and ceiling effects in the symptomatic players subgroup, and only a ceiling effect was found at one time period for the whole group, it is assumed that the asymptomatic respondents of the whole group are the ones to whom the ceiling effect can be attributed. These results are consistent with the study of Schulz et al. [17] and Turgut and Tunay [11], who also observed a marginal ceiling effect for the KJOC questionnaire in asymptomatic overhead athletes. Moreover, due to the absence of clear ceiling and floor effects for the KJOC questionnaire, the discriminatory capacity of this questionnaire is much better compared to the DASH questionnaire. This statement is supported by the observed floor effect of the DASH, showing that a zero score on the DASH questionnaire corresponded with a score range from 70 to 100 on the KJOC questionnaire (Figure 1a–c). A plausible reason for this is that the overhead athletes, and, in this study, baseball pitchers in particular, may experience upper extremity symptoms in their sport-specific environment, whereas activities in daily life can be performed without any problems.

The present study does have some limitations. Firstly, since the KJOC questionnaire can be used to monitor the functional status of the upper extremity and throwing performance in baseball pitching, the questionnaire also has the potential to be used to evaluate the return-to-sport and return-to-competition ability after shoulder and elbow injuries in baseball pitchers. However, this study does not provide information about responsiveness, which would provide valuable information about the ability to detect clinically important changes over time [22]. Previous cross-cultural validation studies showed that the KJOC questionnaire appears to be responsive [8,16], but since this psychometric property may vary between overhead sports populations, it is important to evaluate the responsiveness in the population of interest. Secondly, self-reported questionnaires are known to be at risk for reporting and recall bias. Since the baseball pitchers were asked to complete the questionnaires based on any symptoms in the past 6 months, the risk of reporting and recall bias may exist, affecting the results of this study. This means that the evaluated psychometric properties of the KJOC may be even better when baseball pitchers report about symptoms at the time of completing the questionnaire. However, a previous study showed that participants were able to accurately recall their previous level of functioning with the QuickDASH questionnaire for up to two years [31]; thus, to what extent these biases affected the results of this study is unclear, but is expected be minimal. Lastly, another caveat to be made here is that a minimum of 50 participants is required, according the guidelines set by Terwee et al. [22], to appropriately analyze questionnaires measuring an individual’s health status. However, the number of baseball pitchers that belonged to the symptomatic subgroup is 23, spread over three periods. This relatively small sample size probably arose as only baseball pitchers who had experienced symptoms in the past six months were requested to complete the three questionnaires. The fact that, in the first period, as many as 14 players were in the symptomatic subgroup, while there were six in the second period and seven in the third period, may explain the widespread internal consistency and construct validity scores across the three periods in the symptomatic subgroup. Besides these limitations, the conclusions of this study are not only applicable to a small homogeneous population, but are more widely applicable due to the relatively large age range of the baseball pitchers that participated in this study.

By validating the Dutch version of the KJOC against the Dutch DASH and WOSI, this study can now support the application of the Dutch KJOC in sporting establishments in the Netherlands and other Dutch-speaking states. This means that coaches can provide players who are not proficient in English with an appropriate survey to analyze the functional status of the upper extremity in the Dutch overhead athletes and monitor changes throughout the season, ultimately improving interpersonal communication within teams.

## 5. Conclusions

Overall, this cross-cultural validation study demonstrated that the Dutch KJOC has good internal consistency and construct validity. The Dutch KJOC has no clear floor and ceiling effects and is able to successfully and accurately assess the functional status of the upper extremity in the sport-specific, high-functioning population of talented juvenile Dutch baseball pitchers.

## Figures and Tables

**Figure 1 sports-10-00163-f001:**
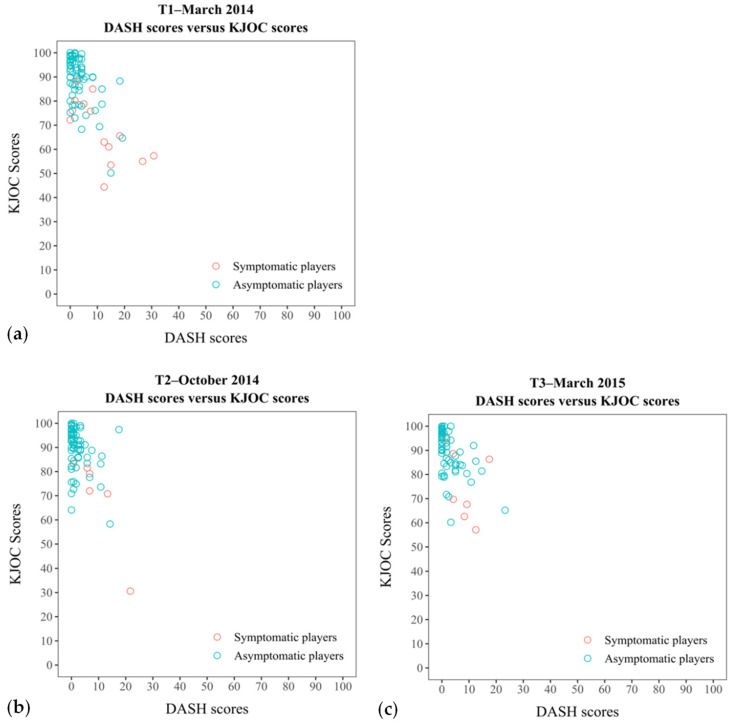
Scatterplot shows the summed KJOC scores plotted against the DASH scores for the first measurement (**a**), second measurement (**b**) and third measurement (**c**) for the asymptomatic players and symptomatic players with upper extremity symptoms.

**Table 1 sports-10-00163-t001:** Participant demographics of the talented juvenile baseball pitchers at the three measurement periods. Data are given as mean (SD).

Period	Participants (*N*)	Age (Years)	Body Height (cm)	Body Weight (kg)	BMI
T1—March 2014	87	14.6 (1.7)	178.0 (11.7)	68.8 (15.3)	21.5 (3.1)
T2—October 2014	79	15.0 (1.7)	179.6 (11.0)	69.4 (15.8)	21.3 (3.3)
T3—March 2015	79	14.9 (1.7)	178.5 (10.6)	69.9 (14.5)	21.7 (3.0)

**Table 2 sports-10-00163-t002:** Mean (SD) of the sum scores for the KJOC, DASH and WOSI questionnaires for the whole group and for the group of pitchers with upper extremity symptoms, where *N* is equal to the number of respondents that filled in the questionnaire.

Period	KJOC	DASH	WOSI
Sum score ranges	0 (worst)–100 (best)	0 (best)–100 (worst)	0 (best)–2100 (worst)
All Players (*N*)			
T1 (87)	86.5 (13.2)	4.3 (6.1)	
T2 (79)	88.9 (11.6)	2.5 (4.3)	
T3 (78)	89.4 (10.6)	2.8 (4.6)	
Symptomatic Players (*N*)			
T1 (14)	68.3 (13.1)	11.1 (9.5)	665 (303)
T2 (6)	78.1 (17.1)	9.2 (7.3)	488 (254)
T3 (7)	83.3 (12.1)	8.2 (5.5)	562 (225)

The symptomatic players who filled out the WOSI questionnaire experienced in any case shoulder symptoms in the last six months.

**Table 3 sports-10-00163-t003:** Cronbach’s alpha values (with corresponding 95% confidence intervals) to assess internal consistency for each of the three measurement periods, where *N* is equal to the number of sampled items in the questionnaire.

Period	Questionnaire	All Players	Players with Upper Extremity Symptoms
T1	KJOC (*N* = 10)	0.83 (0.75, 0.88)	0.62 (0.20, 0.80)
T2	KJOC (*N* = 10)	0.84 (0.71, 0.91)	0.86 (−1.30, 0.94)
T3	KJOC (*N* = 10)	0.84 (0.77, 0.88)	0.82 (0.46, 0.90)

**Table 4 sports-10-00163-t004:** Spearman’s rho values for the correlation between the KJOC, DASH and WOSI questionnaire sum scores to indicate the construct validity of the KJOC questionnaire.

Period	All Players	Players with Upper Extremity Symptoms	
	DASH	DASH	WOSI
T1	−0.59	−0.69	−0.87
T2	−0.47	−0.99	−0.32
T3	−0.67	−0.58	−0.60

The symptomatic players who filled out the WOSI questionnaire experienced in any case shoulder symptoms in the last six months.

**Table 5 sports-10-00163-t005:** Floor and ceiling effects for the KJOC, DASH and WOSI questionnaire sum scores for the whole group and for the group of players with upper extremity symptoms for each measurement period.

	KJOC		DASH		WOSI	
Sum Score Ranges	0 (worst)–100 (best)		0 (best)–100 (worst)		0 (best)–2100 (worst)	
Effect	Floor	Ceiling	Floor	Ceiling	Floor	Ceiling
All Players						
T1	No: 0	No: 9 (10.3%)	Yes: 25 (28.7%)	No: 0		
T2	No: 0	No: 9 (11.4%)	Yes: 33 (41.8%)	No: 0		
T3	No: 0	Yes: 12 (15.2%)	Yes: 38 (48.1%)	No: 0		
Symptomatic players with upper extremity symptoms						
T1	No: 0	No: 0	No: 1 (7.1%)	No: 0	No: 0	No: 0
T2	No: 0	No: 0	No: 0	No: 0	No: 0	No: 0
T3	No: 0	No: 0	No: 0	No: 0	No: 0	No: 0

The symptomatic players who filled out the WOSI questionnaire experienced in any case shoulder symptoms in the last six months; *N* (%) number and percentage of participants obtaining the maximal or minimal score.

## Data Availability

The datasets presented in this study can be found in online repositories. The dataset is available from 24 October 2022. The names of the repository/repositories and accession number(s) can be found below: 10.5281/zenodo.7032703.

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
