# Peer review of "Cross-Cultural Adaption and Validation of the Dutch Version of the Kerlan-Jobe Orthopaedic Clinic Questionnaire in Juvenile Baseball Pitchers"

_sports, 2022, doi:10.3390/sports10110163_

Round 1

Reviewer 1 Report

First of all, I would like to thank the authors for the presented results of their investigation on juvenile baseball pitchers. Also, I would like to thank the editor for the opportunity to review this manuscript.

In a study of talented young Dutch baseball pitchers, the authors of a manuscript titled "Cross-cultural adaptation and validation of the Dutch version of the Kerlan-Jobe Orthopaedic Clinic (KJOC) questionnaire in juvenile baseball pitchers" compare and contrast the results of the Dutch translation of the original English KJOC with those of the valid DASH and WOSI questionnaires. In my opinion, the authors present an interesting topic that falls within the aims and scope of the Sports journal. Although the KJOC questionnaire was developed in English and has been translated into numerous other languages, there has been no study of the Dutch translation of the original English version of the KJOC questionnaire and its validity compared to the validated Dutch versions of the DASH and WOSI questionnaires.

The paper is well structured and easy to follow. I see a wide range in the age of the respondents as the only major vagueness. I would recommend the authors consider this issue to strengthen the paper.

Minor suggestions:

Line 3: I would suggest avoiding the abbreviation in the title.

Lines 15-28: In contrast to the previous comment, I would suggest defining abbreviations when they appear for the first time.

Line 64: I suppose that in brackets should be the number of the reference rather than the year of publication.

Lines 107-127: It makes sense to provide gender identification of the sample.

Line 121: I suppose that in brackets should be the number of the reference rather than the year of publication.

Line 154: The space is missing after 70 and before mm.

Lines 291-293: I am suggesting past tense, show → showed.

Lines 296-299: I am suggesting past tense, are → were.

Lines 307-309: I am suggesting past tense, show → showed.

Lines 311-312: I am suggesting past tense, are → were.

Lines 312-316: I am suggesting past tense, consists → consisted.

Line 372: I suppose that in brackets should be the number of the reference rather than the year of publication.

Author Response

First of all, thank you very much for taking the time to read the manuscript and provide meaningful constructive feedback to improve our work even further!

All minor suggestions have been implemented in the newly uploaded manuscript by using the 'Track Changes' option in microsoft office (word). 

At the end of the limitation section in the discussion, we briefly elaborated on the age range of the respondents to strengthen the paper (lines 392 - 394).

The manuscript has also been re-read by one of our co-authors who is a native English speaker.

Reviewer 2 Report

No advice necessary. The text is very clear and informative. Congratulations!

Author Response

Thank you for very much for taking the time to read the manuscript. Glad to hear that our work turns out to be clear and informative.

The other reviewer requested very minor revisions and these have been implemented in the newly uploaded manuscript.
